# Subnational Gender Inequality and Childhood Immunization: An Ecological Analysis of the Subnational Gender Development Index and DTP Coverage Outcomes across 57 Countries

**DOI:** 10.3390/vaccines10111951

**Published:** 2022-11-18

**Authors:** Nicole E. Johns, Katherine Kirkby, Tracey S. Goodman, Shirin Heidari, Jean Munro, Stephanie Shendale, Ahmad Reza Hosseinpoor

**Affiliations:** 1Department of Data and Analytics, World Health Organization, 20 Avenue Appia, 1211 Geneva, Switzerland; 2Department of Immunization, Vaccines, and Biologicals, World Health Organization, 20 Avenue Appia, 1211 Geneva, Switzerland; 3Gavi, the Vaccine Alliance, Chemin du Pommier 40, Le Grand-Saconnex, 1218 Geneva, Switzerland

**Keywords:** immunization, vaccination, zero-dose children, diphtheria-tetanus-pertussis vaccine, determinants of immunization, health status disparities, gender equity, gender inequality

## Abstract

The role of gender inequality in childhood immunization is an emerging area of focus for global efforts to improve immunization coverage and equity. Recent studies have examined the relationship between gender inequality and childhood immunization at national as well as individual levels; we hypothesize that the demonstrated relationship between greater gender equality and higher immunization coverage will also be evident when examining subnational-level data. We thus conducted an ecological analysis examining the association between the Subnational Gender Development Index (SGDI) and two measures of immunization—zero-dose diphtheria-tetanus-pertussis (DTP) prevalence and 3-dose DTP coverage. Using data from 2010–2019 across 702 subnational regions within 57 countries, we assessed these relationships using fractional logistic regression models, as well as a series of analyses to account for the nested geographies of subnational regions within countries. Subnational regions were dichotomized to higher gender inequality (top quintile of SGDI) and lower gender inequality (lower four quintiles of SGDI). In adjusted models, we find that subnational regions with higher gender inequality (favoring men) are expected to have 5.8 percentage points greater zero-dose prevalence than regions with lower inequality [16.4% (95% confidence interval (CI) 14.5–18.4%) in higher-inequality regions versus 10.6% (95% CI 9.5–11.7%) in lower-inequality regions], and 8.2 percentage points lower DTP3 immunization coverage [71.0% (95% CI 68.3–73.7%) in higher-inequality regions versus 79.2% (95% CI 77.7–80.7%) in lower-inequality regions]. In models accounting for country-level clustering of gender inequality, the magnitude and strength of associations are reduced somewhat, but remain statistically significant in the hypothesized direction. In conjunction with published work demonstrating meaningful associations between greater gender equality and better childhood immunization outcomes in individual- and country-level analyses, these findings lend further strength to calls for efforts towards greater gender equality to improve childhood immunization and child health outcomes broadly.

## 1. Introduction

Gender inequality is increasingly recognized as a key determinant of childhood immunization coverage and health equity [1,2,3,4]. Gender-related barriers to immunization have been shown to operate at the individual, interpersonal, community, and broader socio-structural levels [2]. These include barriers faced by (frequently women) caregivers, such as lower health education and literacy, travel restriction, and limited household decision-making influence; by health workers delivering services (who are disproportionately women), including gender pay gap, workplace harassment and inequitable exposures to health risks; and by policy-makers (where women are frequently under-represented), who enact laws and guidelines which may amplify or reinforce gender inequities [2,5]. Several recent studies have examined the relationship between childhood immunization coverage and measures of gender inequality empirically, at the individual [6,7] and national [8,9] levels. These studies consistently find significant and meaningful associations between greater gender inequality and lower immunization coverage.

Existing individual-level analyses use the survey-based women’s empowerment (SWPER) index, a three-dimensional measure of women’s empowerment comparable across time and geographies [10]. These studies find that children of women with greater empowerment (as measured by social independence [including such items as schooling attainment and access to information], decision-making control, and attitudes towards violence) were more likely to have received three doses of the combined diphtheria-tetanus-pertussis (DTP) vaccine and less likely to have received zero doses of DTP than children of women with lower empowerment [6,7]. Individual-level analyses have several advantages: mothers are frequently caregivers for their child, and their experiences are proximally related to their child’s outcomes; confounding mother- and child-level information known to be associated with immunization coverage could be accounted for, including mother’s education and child birth order; and unlike aggregated analyses, these methods can avoid the ecological fallacy and account for individual variation. However, individual measures of empowerment do not take into account broader gender norms, policies, and social climates that may contribute to gender inequality. Furthermore, it is difficult to assess empowerment or gender equity at the individual level given existing measures.

National level analyses that have examined gender barriers and immunization outcomes similarly find that countries with lower gender inequality have higher rates of DTP3 coverage and lower zero-dose DTP prevalence [8,9]. The advantages of national analyses include: readily available data and the ability to examine large numbers of geographies; standard measures of inequality that are comparable across countries and time; and the fact that national averages capture the broader state of women in a society, as laws, economics, health systems, and education are often determined and implemented at the national level. However, these analyses fail to account for individual variation and may reflect averages which obscure more important within-country inequality. They also fail to capture community factors at the subnational level, where there may be significant differences in regional policies or implementation of national practices and priorities.

Our current analysis expands on this previous work and fills an important gap by utilizing subnational data to examine the association between gender inequality and childhood immunization at the subnational region level. Although subnational analyses also cannot capture all levels at which gender inequality may affect child immunization, they do bridge the gap between existing national and individual level information. Subnational units may be particularly relevant for laws, health systems, government or nonprofit initiatives, as well as geographic variation in education, religion, wealth, industry, and other factors which may be associated with both gender equity and childhood immunization. Specifically, in this manuscript we test the hypothesis that the subnational gender development index will be associated with zero-dose DTP prevalence and DTP3 coverage at the subnational level.

## 2. Materials and Methods

### 2.1. Indicators and Data Sources

The data used in this study include up to 10 years of subnational region estimates of childhood immunization, indicators of gender inequality, and other demographic, economic, and social characteristics. Data were available for 702 subnational regions across 57 countries. We included the 10 most recent years of available data (2010–2019); all region years where estimates for subnational gender development and immunization outcomes were available were included, for a total of 1066 total region years of data. 

#### 2.1.1. Immunization Outcomes

We examined two outcomes based on subnational coverage of the DTP vaccine. First, the prevalence of zero-dose children (zero-dose DTP), defined as the percentage of surviving one-year old children in a subnational region who have not received the first dose of the DTP vaccine series. This indicator is a proxy for children who have missed immunization services entirely. Second, the prevalence of DTP3 immunization (DTP3), the percentage of surviving one-year old children in a subnational region who have received three doses of DTP vaccine. This indicator is a proxy for children who have accessed the full series of basic immunizations. Together, these are frequently used indicators of child health more broadly as they reflect regular and timely interaction with health services (DTP3) and health equity (zero-dose DTP) [11,12,13].

These estimates are derived from Demographic and Health Surveys (DHS) Program data, which uses a rigorous survey design to create representative samples at the subnational level. Substantial detail on the study design and methodology of the DHS has been published elsewhere [14].

#### 2.1.2. Factors Associated with Immunization Coverage

We examined variables selected a priori based on prior national-level analyses, to make findings as directly comparable as possible [9]. These factors were chosen to account for demand and supply side factors that influence vaccination and might confound the association between immunization and gender inequality [15,16,17,18,19]. These included subnational estimates of percent of population under 15 years of age, percent of population living in urban areas, and a number of human development indicators (described below). We also utilized national estimates of average annual rate of population change; estimates corresponding to study subnational regions were not readily available.

To capture human development in adjusted models, we utilized the subnational human development index (SHDI). The SHDI is a summary measure of development in three dimensions, namely education, health, and standard of living, with an index normalized between 0 and 1 created for each dimension [20]. The education index based on mean expected years of schooling for children and mean years of schooling for adults ages 25 years and older, the health index is based on life expectancy at birth, and the standard of living index is based on gross national income per capita (2017 purchasing power parities [PPP] in USD). We utilized the three dimension-specific indices in analyses. Each of these indices are calculated both for the total population, as well as disaggregated by sex. All human development indicators were available at the subnational level.

#### 2.1.3. Gender Inequality

Gender inequality was measured using the subnational gender development index (SGDI) [20,21]. The SGDI is the only readily publicly available metric of gender inequality available at the subnational level which is comparable across geographies and time.

SGDI captures gender inequalities in achievement in the three dimensions of development captured by the SHDI (items detailed above). The SGDI is the ratio of SHDI among men to SHDI among women within a subnational region; additional detail regarding the SGDI is published elsewhere [20]. We include both SGDI (the ratio of development between women and men) as well as the SHDI (the overall level of development) in adjusted models.

SGDI values below 1 indicate higher human development among men than women, a value equal to 1 indicates equality, and values above 1 indicate higher development among women than men. We created a binary analysis variable for SGDI based on quintiles of its sample distribution, dichotomized to higher gender inequality favoring men (highest quintile) versus lower gender inequality (quintiles 2–5). In analyses limited to the most recent year of data, we recreated the binary variable based on quintiles of the most recent year sample distribution. We present summary statistics for the continuous SGDI measure, but analyzed SGDI as a binary measure (higher versus lower gender inequality) in regression analyses for ease of interpretation.

### 2.2. Data Sources

All subnational estimates of outcomes, gender inequality, human development, and demographic characteristics came from the Global Data Lab [22]. Though the Global Data Lab produces SHDI estimates for subnational regions in 161 countries for all years from 1990–2019, we utilized only those country years in which a DHS survey was conducted, as subnational vaccination coverage was only available for these years. As a result, all data in this study is derived from DHS survey-weighted estimates and do not rely on interpolation. Full details on data sources for the indicators compiled, calculated, and distributed by Global Data Lab have been published elsewhere [20]. Estimates of national average annual rate of population change came from the World Development Indicators [23].

Table 1 presents a summary of indicators.

### 2.3. Analyses

We present descriptive statistics, bivariate comparisons of immunization outcomes and SGDI, and unadjusted outcome distributions by SGDI, for the most recent year of data available for each subnational region. We then present regression analyses to examine the association between childhood immunization and gender inequality using the full 10-year dataset. All region years with available data were included in analyses. All models were conducted using fractional logit specifications, as the outcomes are proportions with values between 0 and 1 [24,25].

Models were estimated with SGDI as a binary variable equal to 1 if subnational regions were in the highest gender inequality quintile, and 0 if regions were in any of the four lower inequality quintiles.

For each immunization outcome, we first estimated the unadjusted association between the outcome and SGDI, without controlling for any other factors. We then conducted adjusted analyses, including controls for annual population growth and age structure, percentage of urban population, and the three individual dimensional indices of the SHDI (health, education, and income).

Unadjusted and adjusted models accounted for non-parametric time trends via year fixed effects, and were estimated with standard errors clustered at the subnational region level.

To account for the geographically clustered nature of subnational regions within countries, we also conducted a series of analyses accounting for country-level clustering.

First, we replicated the adjusted fractional logistic regression as described above with the addition of a covariate which was the country-year average zero-dose DTP prevalence or DTP3 coverage.Second, we retained only the most recent year of available data for each subnational region, and conducted the same adjusted fractional logistic regression, but with the clustered standard errors based on country, rather than region.Third, we included all available data but used a multi-level mixed effects linear regression approach, using nested random effects of subnational region within country, with covariate fixed effects as defined by the adjusted model above. For these models, we specified random intercepts for both country and region, and random slopes for region, with an identity variance-covariance structure; these specifications were selected based on model performance as assessed by AIC and BIC.Fourth, we replicated the mixed-effects linear regression approach using the most recent year of available data for each subnational region, and including only random intercepts for country.

Statistical significance was set at *p* < 0.05 for all comparisons including adjusted odds ratios (AORs); 95% confidence intervals (CIs) are reported throughout. All analyses were conducted using STATA 16.1 [26].

## 3. Results

### 3.1. Descriptive Analyses

In the most-recent-year sample, where each observation is one region, the mean value of SGDI was 0.90, ranging from a low of 0.51 to a high of 1.09. This mean value below 1 indicates that, overall, human development was lower among women than men in the analyzed subnational regions. Distributions of the SGDI for the pooled 10-year (Figure 1a) and most-recent-year (Figure 1b) samples are shown in Figure 1.

In unadjusted comparisons, higher gender inequality was associated with higher prevalence of zero-dose DTP and lower DTP3 immunization coverage (Table 2, Figure 2 and Figure 3). Examining the most recent year of available data sample, subnational regions with higher gender inequality (favoring men) as measured by the SGDI had 13.4 percentage points greater zero-dose prevalence (18.2% vs. 4.8%), and 21.6 percentage points lower DTP3 immunization coverage (86.0% vs. 64.4%) than regions with lower inequality.

### 3.2. Regression Analyses

Higher inequality was significantly associated with lower zero-dose prevalence and higher DTP3 coverage in unadjusted and adjusted fractional logistic regression analyses (Table 3). In subnational regions with higher gender inequality, zero-dose prevalence odds were 1.7 times higher (AOR = 1.74, 95% CI: 1.38–2.19) compared to subnational regions with lower inequality. Consistently, the odds of DTP3 coverage were 39% lower (AOR = 0.61, 95% CI: 0.51–0.75) in regions with higher gender inequality relative to regions with lower inequality.

We also estimated the average marginal effects of SGDI to indicate the average percentage point change in the outcome variable (zero-dose DTP or DTP3 coverage) by higher versus lower gender inequality (See Figure 4). A subnational region with higher inequality (favoring men) is expected to have 5.8 percentage points higher prevalence of zero-dose DTP relative to a region with lower inequality, increasing from 10.6% (95% CI 9.5–11.7%) for regions with lower inequality to 16.4% (95% CI 14.5–18.4%) for regions with higher inequality. A subnational region with higher gender inequality is expected to have 8.2 percentage points lower coverage of DTP3 immunization than a region with lower gender inequality, dropping from 79.2% (95% CI 77.7–80.7%) for regions with lower inequality to 71.0% (95% CI 68.3–73.7%) for regions with higher inequality.

#### Models Accounting for Country-Level Clustering

Consideration of country-level clustering reduced the observed associations between subnational gender inequality and immunization coverage outcomes. In the model additionally controlling for the average zero-dose prevalence or DTP3 coverage for the corresponding country-year, we find a significant association between gender inequality and both zero-dose DTP prevalence and DTP3 coverage. In the model limited to the most recent year of data available for each subnational region and clustering standard errors by country, we do not observe a significant association between gender inequality and immunization outcomes. In multilevel linear regression models accounting for nested random effects of subnational regions within country, we find significant associations between gender inequality and both zero-dose DTP prevalence and DTP3 coverage. Findings are similar when limited to the most recent year of data, utilizing a linear regression model with country random effects. To more directly compare findings between models, we present predicted marginal effects of higher versus lower gender inequality, e.g., the predicted percentage point difference in coverage between subnational regions with higher gender inequality compared to those with lower gender inequality (see Table 4). We first present the adjusted model that does not account for country clustering, as well as the four models discussed above. Though the direction of association remains constant across models, the magnitude and strength of association is reduced for the models that take into account country-level clustering.

## 4. Discussion

Findings from this study of 702 subnational regions across 57 countries suggest that greater gender equality, as measured by the SGDI, is associated with positive childhood immunization outcomes—higher DTP3 coverage and lower zero-dose prevalence. We find that, after adjustment, a subnational region with higher gender equality is expected to have 5.8 percentage points lower prevalence of zero-dose DTP and 8.2 percentage points higher coverage of DTP3 than a region with lower gender equality. To put this coverage difference in context, it took more than 10 years of concerted effort for global DTP3 coverage to improve by 8 percentage points—DTP3 coverage globally increased from 78% in 2006 to 86% in 2019 (prior to COVID-19-related declines) [27].

These findings align with prior work examining gender inequality and childhood outcomes, including child mortality and immunization coverage, using different analytic approaches including alternate measures of gender inequality and national or individual units of analysis [6,7,8,9,28,29,30]. These studies consistently find that gender equality, and the related construct of women’s empowerment, are associated with improved immunization coverage, decreased child mortality, and other positive child health outcomes. Existing work has also demonstrated substantial subnational inequality in immunization, highlighting the relevance of subnational policies and outreach efforts, as well as intra-country variations in immunization access and resources [31,32]. Our study builds on this existing literature to demonstrate that within-country variation in gender inequality is associated with immunization coverage at the subnational level, and suggests that gender inequality may be one of many drivers of subnational inequalities in coverage.

Compared to national analyses, we find an even stronger association between immunization and subnational gender inequality [9]. For example, the same adjusted regressions suggests that at the national level, countries with higher gender equality have 4.6 percentage points higher DTP3 coverage than countries with lower gender equality, while we find that subnational regions with higher gender equality had 8.2 percentage points higher DTP3 coverage than subnational regions with lower gender equality. This larger (and statistically stronger) association highlights the importance of within-country variation in determinants of immunization. Nonetheless, we do find that the magnitude of these associations is reduced somewhat when we take into account the clustering of subnational regions within countries. This reduction in effect size suggests that national-level factors remain important and meaningful predictors of immunization.

Reaching zero-dose and under-immunized children means reaching the communities they are a part of; these ‘missed communities’ are not only a heightened risk for disease outbreaks, but often also suffer from a lack of basic services and face entrenched socio-economic marginalization [33]. Better understanding the drivers of subnational inequalities—such as subnational differences in gender inequality—can enable targeted and tailored approaches to improve not only gender equality, but also reach these missed communities to improve immunization coverage and equity.

Findings from this study should be viewed in light of its limitations. Firstly, these are ecological analyses, and hence does not imply causation. However, taken together, the consistent association between gender equality and better childhood immunization coverage across a range of individual, national, and subnational analyses lend strength to the assertion that gender inequality is a key determinant of immunization coverage and equity. Second, these data are available for low- and middle-income countries; high-income countries, which likely have stronger health systems, and other countries without available data may or may not exhibit the same patterns of association. Third, while these findings demonstrate an association between gender inequality and immunization coverage, they do not elucidate the pathways through which that association may be causal. Qualitative work is needed to better understand the contextual pathways through which restrictive gender norms and gender-related barriers hamper immunization efforts.

A growing body of evidence on gender as a determinant of health examines the ways in which gender inequality influences decision-making about health services, access to and affordability of health services, limitations on mobility and decision-making, and provider attitudes, among others [4,34,35]. Further work is needed to understand the ways in which interventions may operate across these pathways, and understand which interventions are effective in addressing and circumventing gender-related barriers to immunization. Addressing these factors in order to improve child immunization coverage and equity are strategic priorities of major international immunization initiatives including the Immunization Agenda 2030 (IA2030) and the Gavi Phase 5 strategy [36,37]. Ensuring gender transformative approaches and efforts to improve gender equality will not only have a benefit for childhood immunization coverage, but better health outcomes for all.

## 5. Conclusions

Our study of 702 subnational regions across 57 countries suggests that gender equality is positively associated with childhood immunization coverage at the subnational level. These findings fill a gap in the existing literature and strengthen findings of individual- and national-level analyses, which collectively show a robust and meaningful association between gender inequality and immunization coverage outcomes. Multi-sectoral gender-responsive and gender-transformative approaches are needed to ensure improvements in immunization coverage and equity.

## Figures and Tables

**Figure 1 vaccines-10-01951-f001:**
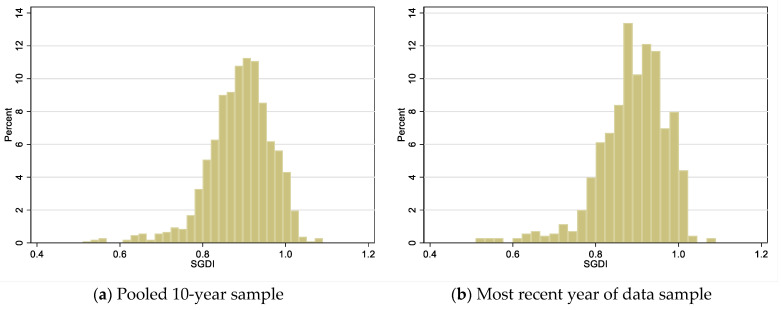
Distribution of Subnational Gender Development Index score.

**Figure 2 vaccines-10-01951-f002:**
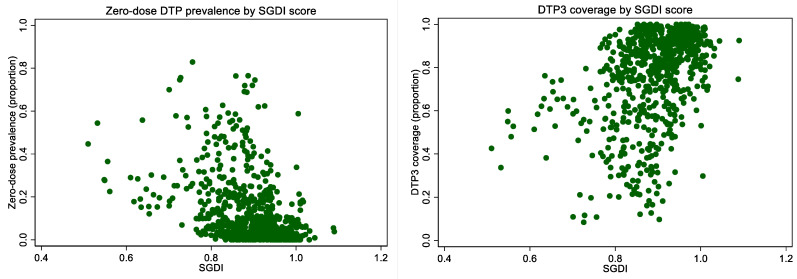
Prevalence of zero-dose DTP and DTP3 immunization coverage by continuous SGDI score, most recent year of available data.

**Figure 3 vaccines-10-01951-f003:**
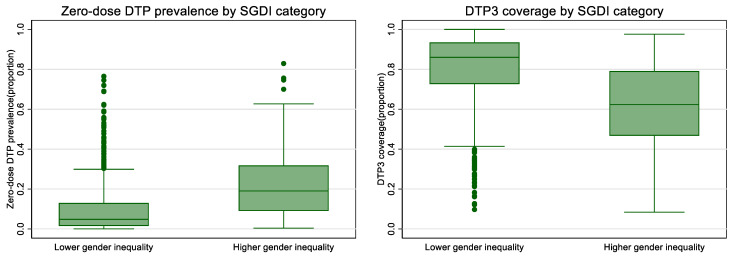
Prevalence of zero-dose DTP and DTP3 immunization coverage by SGDI category, most recent year of available data. Note that boxes show 25–75th percentile values, with 50th percentile (median) line inside box. Single dots are outlier values.

**Figure 4 vaccines-10-01951-f004:**
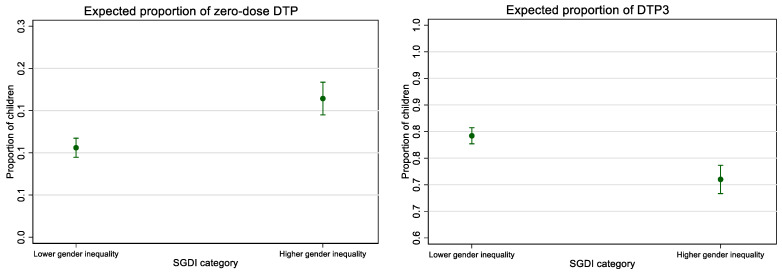
Adjusted * expected proportions of zero-dose DTP and DTP3 immunization coverage by SGDI category, 702 subnational regions across 57 countries, 2010–2019. Estimated proportions are adjusted for annual population growth and age structure (measured as the percentage of the population under 15 years of age), percentage of urban population, and the three individual dimensional indices of the SHDI (health index, education index, and income index).

**Table 1 vaccines-10-01951-t001:** Measures.

Category	Indicator
Outcomes	Zero-dose DTP prevalence
	DTP3 immunization coverage
Gender inequality	Subnational gender development index (SGDI)
Demographic/geographic characteristics	Average annual rate of population change (%) *
	Population <15 years (%)
	Urban population (%)
Human development	Subnational health index (0 to 1)
	Subnational education index (0 to 1)
	Subnational income index (0 to 1)

* All indicators at the subnational level with the exception of annual rate of population change, which is assessed at the national level due to data availability.

**Table 2 vaccines-10-01951-t002:** Prevalence of zero-dose DTP and DTP3 immunization coverage by SGDI category, most recent year of available data.

	Zero-Dose DTP (%)			DTP3 Immunization Coverage (%)			
	Median	Min	Max	Median	Min	Max	*N*
High gender inequality	18.2	0	96.6	64.4	2.6	98.1	*214*
Medium/low/negligible gender inequality	4.8	0	81.2	86.0	9.7	100	*852*
*p*-value	<0.001			<0.001			

**Table 3 vaccines-10-01951-t003:** Odds ratios for zero-dose DTP prevalence and DTP3 immunization coverage by SGDI category (702 subnational regions across 57 countries, 2010–2019).

	Unadjusted	Adjusted
**Zero-dose children**		
High gender inequality	2.637 ***	1.742 ***
95% CI	(2.122–3.275)	(1.384–2.193)
**DTP3 immunization coverage**		
High gender inequality	0.437 ***	0.614 ***
95% CI	(0.364–0.524)	(0.505–0.746)

*** *p* < 0.001.

**Table 4 vaccines-10-01951-t004:** Predicted marginal effects [percentage point difference] for zero-dose DTP prevalence and DTP3 immunization coverage by SGDI category (702 subnational regions across 57 countries, 2010–2019).

	No Country Consideration (Fractional Logistic Model, Full Sample)	Fractional Logistic Model, Plus Country-Year Average Coverage	Fractional Logistic Model, Most Recent Year of Data Only, Country Clustered Standard Errors	Mixed Effects Linear Regression Model, Nested Random Effects	Mixed Effects Linear Regression Model, Most Recent Year of Data Only, Country Random Effect
*N*	*1066*	*1066*	*702*	*1066*	*702*
**Zero-dose children**					
High gender inequality	5.83 **	3.31 **	3.48	3.64 *	4.16 *
95% CI	(3.26–8.39)	(1.82–4.80)	(−1.04–8.00)	(1.36–5.91)	(1.57–6.75)
**DTP3 immunization coverage**					
High gender inequality	−8.20 **	−4.07 **	−5.16	−4.22 *	−5.30 *
95% CI	(−11.64 to −4.77)	(−6.06 to −2.09)	(−11.69–1.38)	(−7.00 to −1.45)	(−8.48 to −2.12)

* *p* < 0.01; ** *p* < 0.001.

## Data Availability

Publicly available datasets were analyzed in this study. These data can be downloaded from the following locations: https://globaldatalab.org/ and https://databank.worldbank.org/source/world-development-indicators (both accessed on 30 August 2022).

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
