# Peer review of "Subnational Gender Inequality and Childhood Immunization: An Ecological Analysis of the Subnational Gender Development Index and DTP Coverage Outcomes across 57 Countries"

_vaccines, 2022, doi:10.3390/vaccines10111951_

Round 1

Reviewer 1 Report

The authors are addressing a relevant topic and a provided a well-designed study and well written manuscript. Some comments to improve the article are listed below:

Avoid repetition in the introduction and methods, e.g. concerning the meaning of "dosing" terms. Introduction is quite lengthy. Several parts could be better used in the discussion. Methods is also rather long (consider moving explanations).

Please provide a critical assessment of the quality of the datasources (completeness, validity, ...).

Discussion is rather poor. Please discuss your findings better in relation to other studies. The scarcity of referred studies in the discussion (5 out of 31...) is troublesome. E.g. : line 317 "These findings align with prior work examining gender inequality and childhood immunisation using different analytic approaches..." requires proof and references.

Author Response

Reviewer 1

The authors are addressing a relevant topic and a provided a well-designed study and well written manuscript. Some comments to improve the article are listed below:

We thank the reviewer for their feedback and for their assistance in strengthening the manuscript.

  • Avoid repetition in the introduction and methods, e.g. concerning the meaning of "dosing" terms. Introduction is quite lengthy. Several parts could be better used in the discussion. Methods is also rather long (consider moving explanations).

We have reduced text in the introduction as well as moved text to the methods or discussion where appropriate. We have shortened the first paragraph and removed most of the second (where there was mention of dosing) and have tried to reduce wordiness throughout. We feel it important to retain the context of recent studies which this manuscript builds upon (currently Introduction paragraphs 2 & 3), as those studies and the noted strengths and weaknesses of each motivated the current study. We have also reduced wordiness in the methods section and removed unnecessary detail, but have retained most content to ensure reproducibility.

  • Please provide a critical assessment of the quality of the datasources (completeness, validity, ...).

We have added additional information regarding the data to Methods subsection 2.2 Data sources, lines 177-182. Though Global Data Lab produces estimates for many more region-years, we utilized only region-years in which DHS surveys were implemented, so all data is complete and derived from actual observation (e.g. we do not utilize any projections or interpolated data). DHS survey data is discussed in lines 122-125 and the authors do not feel that further discussion of DHS survey data is needed.

  • Discussion is rather poor. Please discuss your findings better in relation to other studies. The scarcity of referred studies in the discussion (5 out of 31...) is troublesome. E.g. : line 317 "These findings align with prior work examining gender inequality and childhood immunisation using different analytic approaches..." requires proof and references.

Thank you for the opportunity to expand and strengthen the discussion. To strengthen the specific statement mentioned, we have added the four works described in the introduction as well as three of the most relevant other examples of such analyses. We have also restructured the discussion to hopefully present a more logical flow, reworded for clarity, and added additional citations throughout.

Reviewer 2 Report

The paper is well presented. However, there are some minor changes:

Please state Figure 1 a and b in the context. 

Please explain the number "1066" in Table 4. In the material and method section 2.1 states"Data were available for 702 subnational regions..."

Please check the line space from line 201 to line 215.

Please check the reference number format in lines 155 and 173 for" [22]", line 322 "[30,31]".

please check the reference : line 443 ref:29.

Author Response

Reviewer 2

The paper is well presented. However, there are some minor changes:

We thank the reviewer for their feedback and for their assistance in strengthening the manuscript.

  • Please state Figure 1 a and b in the context. 

Reference to the sub-figures has been added in lines 237-238 to clarify.

  • Please explain the number "1066" in Table 4. In the material and method section 2.1 states, "Data were available for 702 subnational regions..."

You are correct that we have data for 702 subnational regions, however, we have data available for multiple years for some regions, for a larger number of observations in the pooled 10 year data sample. Lines 107-109 now clarify: “We included the 10 most recent years of available data (2010-2019); all region-years where estimates for subnational gender development and immunization outcomes were available were included, for a total of 1,066 total region-years of data.”

  • Please check the line space from line 201 to line 215.

Thank you for noting this – we have fixed the line spacing for the bulleted text. [Note to editors – the format change is not shown in track changes but was completed].

  • Please check the reference number format in lines 155 and 173 for" [22]", line 322 "[30,31]".

We have fixed the formatting of these references to match others. [Again, note to editors that this formatting change is not reflected in track changes but was completed].

  • Please check the reference: line 443 ref:29.

We have modified the title for Reference 29.